# Unraveling 5f-6d hybridization in uranium compounds via spin-resolved L-edge spectroscopy

R.D. dos Reis[1,2], L.S.I. Veiga[1,2], C.A. Escanhoela Jr.[1], J.C. Lang[3], Y. Joly[4,5], F.G. Gandra[2], D. Haskel[3] & N.M. Souza-Neto[1,3]

The multifaceted character of 5f electrons in actinide materials, from localized to itinerant and in between, together with their complex interactions with 6d and other conduction electron states, has thwarted efforts for fully understanding this class of compounds. While theoretical efforts abound, direct experimental probes of relevant electronic states and their hybridization are limited. Here we exploit the presence of sizable quadrupolar and dipolar contributions in the uranium $L_3$-edge X-ray absorption cross section to provide unique information on the extent of spin-polarized hybridization between 5f and 6d electronic states by means of X-ray magnetic circular dichroism. As a result, we show how this 5f-6d hybridization regulates the magnetism of each sublattice in $UCu_2Si_2$ and $UMn_2Si_2$ compounds, demonstrating the potentiality of this methodology to investigate a plethora of magnetic actinide compounds.

[1] Brazilian Synchrotron Light Laboratory (LNLS), Brazilian Center for Research in Energy and Materials (CNPEM), Campinas, SP 13083-970, Brazil. [2] Instituto de Fisica Gleb Wataghin, Universidade Estadual de Campinas (UNICAMP), Campinas, SP 13083-970, Brazil. [3] Advanced Photon Source, Argonne National Laboratory, Argonne, IL 60439, USA. [4] Univ. Grenoble Alpes, Inst. NEEL, F-38042 Grenoble, France. [5] CNRS, Inst. NEEL, F-38042 Grenoble, France. Correspondence and requests for materials should be addressed to N.M.S.-N. (email: narcizo.souza@lnls.br)

The extent of $f$-state electronic hybridization with valence/conduction band states is key in defining the physical properties of rare earths and actinide compounds. Materials that defy straightforward description in terms of either localized or itinerant electron behavior are a longstanding challenge to understanding novel electronic matter. The actinide materials are at the heart of this debate because the $5f$ electrons are intermediate between localized and delocalized. In those two limiting cases, well-defined methodologies exist that account for their magnetic behavior such as Hund's Rules, crystal-field theory, and quenched angular momentum theory. In uranium compounds, although the degree of $f$-electronic localization is widely recognized as the dominant factor in determining the structural, magnetic and electronic properties, experimental methods for probing the $5f$, $6d$ states and their hybridization have generally failed. For example, the extent of $5f$-$6d$ hybridization is essential to address some open questions on the itinerant electron behavior in UTe[1], UGe$_2$[2] and the recent interpretation of hastatic order in the heavy-fermion compound URu$_2$Si$_2$[3]. Although we focus here on magnetic properties[4], the $f$-$d$ hybridization also affects other phenomena such as oxidation states[5, 6], electronic structure[7, 8], pressure induced changes[6, 9–12], and the character of bonding[13], among others. A few resonant and non-resonant high-resolution X-ray spectroscopy techniques[1, 5, 7–9, 14] have recently been proposed to study the electronic structure of actinide compounds showing decisive results in some cases. However, a method to directly and selectively probe the electronic $5f$, $6d$ states and their hybridization has not been available yet, which, in turn, is crucial for a comprehensive understanding of the unconventional mechanisms that regulate the physics of $5f$ electrons in actinides. In addition, with the intrinsic difficulties handling actinide elements due to their toxic and radioactive nature, theoretical work on actinide compounds has been much more extensive than experimental work[3, 13, 15, 16]. Therefore the need for a technique capable of directly probing the relevant electronic states, as well as testing theoretical predictions, is abundantly clear.

It is well known that $5f$ states of actinides can be directly probed by X-ray absorption spectroscopy at $M_{4,5}$ ($3d{\rightarrow}5f$) or $N_{4,5}$ ($4d{\rightarrow}5f$) transitions allowing determination of orbital and spin magnetic moment via X-ray magnetic circular dichroism (XMCD) sum rules[17, 18]. But information on the $6d$ states is absent in these measurements. By using $L_{2,3}$-edges spectra instead, which involve $2p{\rightarrow}6d$ and $2p{\rightarrow}5f$ transitions in the dipolar and quadrupolar channels, respectively[11, 19, 20], information on both $5f$ and $6d$ states (and their hybridization) can in principle be obtained. However, experimental difficulties such as the inefficiency to produce circular polarization and the significatly low amplitude signals, ~1/50 when compared to the ones at $M_{4,5}$-edges, have thwarted efforts to perform careful $L_{2,3}$ XMCD studies on uranium or any other actinide compound. We note that the use of magneto-optical sum rules to separately determine the orbital and spin magnetic moments from XMCD spectra as proposed by P. Carra et al.[21] is a unique tool. However, these sum rules normally cannot be directly applied for the case of L-edges of rare earths and actinides due to the presence of $5f$ quadrupolar contributions in the XMCD spectra and the influence of the $5f$-$6d$ hybridization on the spatial extent of $6d$ orbitals. While some ways to overcome this have been proposed[22], it relies on the inclusion of spin-asymmetry parameters and the combination of the experimental data with theoretical calculations, in addition to the correct extraction of the $5f$ quadrupolar contribution from the spectra.

Here we report on XMCD in the uranium $L_3$-edge (17 keV) for UT$_2$Si$_2$ (T = Cu, Mn) compounds. We demonstrate this technique offers a unique opportunity to directly probe the magnetism

of actinides compounds and also address the question of electronic hybridization between $5f$ and $6d$ states. By analyzing the relative amplitudes of (sizable) quadrupolar and dipolar contributions to the XMCD signal at U-$L_3$ edges we describe the individual content of $6d$ and $5f$ spin-polarized states as well as $6d$-$5f$ electronic hybridization in prototypical uranium compounds. The bulk-sensitivity of the method at this relatively high energy also allows investigating these electronic effects as a function of applied pressure.

## Results

**$5f$/$6d$ hybridization in UCu$_2$Si$_2$ and UMn$_2$Si$_2$.** We have selected UCu$_2$Si$_2$ and UMn$_2$Si$_2$ compounds to benchmark this methodology since their family of intermetallics (UT$_2$X$_2$ (1:2:2) with T = transition metal and X = Si or Ge) presents a wide variety of electronic and magnetic properties, often times argued to be due to the relationship between magnetism and the hybridization effects among uranium $5f$ and $3d$/$4d$/$5d$ states. While much attention has been given to UCu$_2$Si$_2$[23, 24], only a few studies on UMn$_2$Si$_2$ are reported so far[25, 26]. The UCu$_2$Si$_2$ compound is reported to show dual itinerant/localized character of the $5f$ electrons being both a strongly anisotropic ferromagnet, with ordering temperature of 103 K, and exhibiting the Kondo lattice behavior. In this picture the ferromagnetic order comes from the part of the U $5f$ highly localized electrons giving rise to the high $T_C$ value. On the other hand, UMn$_2$Si$_2$ orders ferromagnetically below 377 K for the Mn sublattice and below about 100 K for the U sublattice[26]. The electronic structure of this series is characterized by the relative energy positions of the $d$ states of a given transition metal atom compared with those of the uranium $5f$ states. While in the Mn-compound the $3d$ and $5f$ states are closer in energy and the $f$-$d$ hybridization is considered to be the largest among the family of 1:2:2 silicates, in the Cu compound the $3d$ and $5f$ states are most distant and therefore the hybridization is almost negligible[25].

Uranium $L_3$ edge (17.166 keV) X-ray absorption near edge structure (XANES) and XMCD measurements on UCu$_2$Si$_2$ and UMn$_2$Si$_2$ compounds are presented in Fig. 1a, b. Two well-defined peaks are clearly present in both compounds. Similarly to rare earths[11, 19, 20, 27], we relate these spectral features in the XMCD signal to a dipolar ($6d$) contribution (in the high-energy peak) and overlapping quadrupolar ($5f$) and dipolar ($6d$) contributions as a result of $5f$/$6d$ hybridization (low energy peak; see Fig. 1b). This interpretation is also supported by first principle calculations of the XMCD spectra which reproduce well the experimental features and differences between the compounds, as shown in Fig. 1c–e. The theoretical spectra in Fig. 1d,e show that while the quadrupolar contribution (green curve) which essentially probes $5f$ orbitals presents only one peak, two peaks are observed due to the dipolar term (blue curve). While the first dipolar peak at low energy arises from the sizable $5f$-$6d$ hybridization, the second peak comes exclusively from $6d$ unoccupied orbitals. (See Supplementary Note 3 for calculations of the density of states using the LDA + U approach). The large $5f$-quadrupolar term at the $L_3$ edge relies on the fact that the X-ray photon wavelength ($\lambda$=0.722 Å in this case) is comparable to the size of the electron orbit[20]. In other words, the high energy of the actinides L-edges (16–25 keV) enhances the quadrupolar contribution from the transition Hamiltonian operator when compared to lower energy L-edges of rare earths[11, 19, 27] (5–10 keV). In addition, considering that the spin–orbit coupling and presence of a sizeable orbital moment plays a crucial role on the appearance of a quadrupolar contribution[20], we can envision that all magnetic actinide materials may present sizeable quadrupolar contributions in their $L_3$-edge XMCD spectra. Even materials

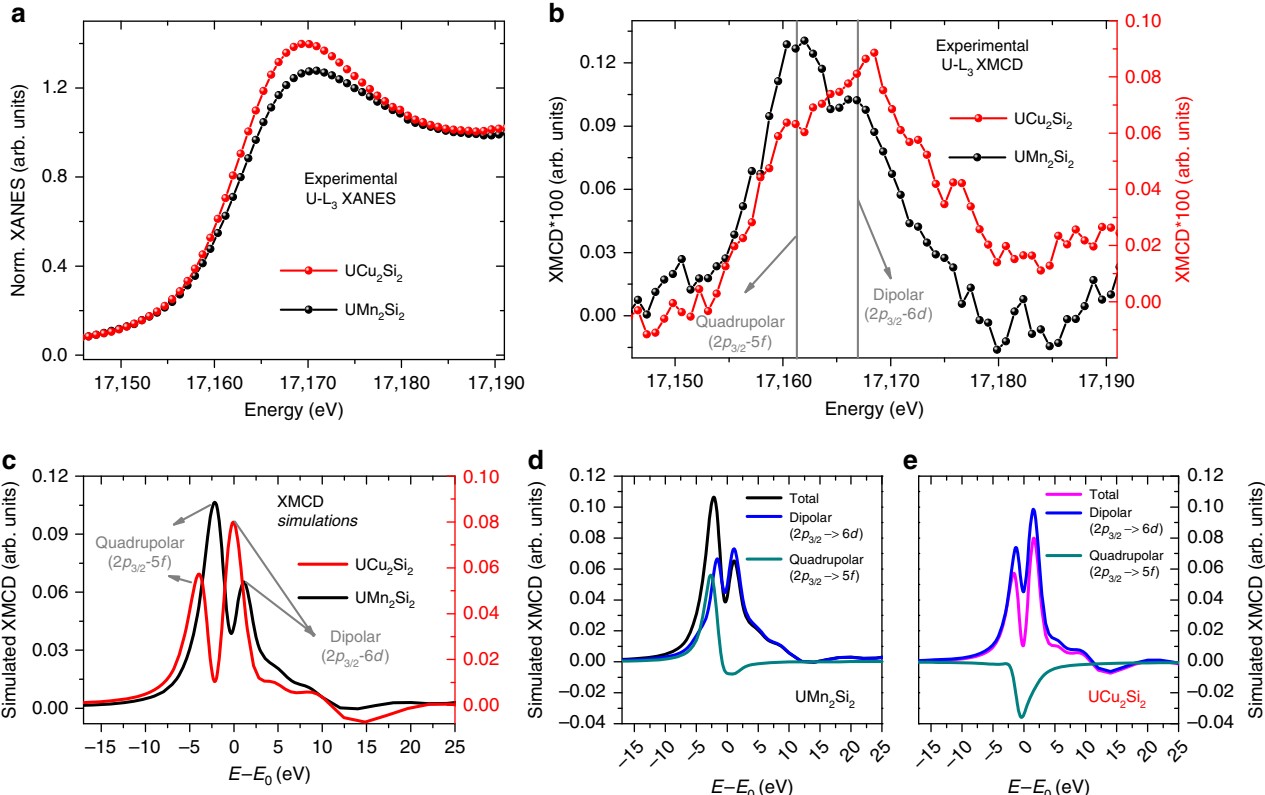

**Fig. 1** Uranium $L_3$ edge spin-dependent X-ray absorption spectroscopy. **a** X-ray Absorption Near Edge Structure (XANES) and **b** X-ray Magnetic Circular Dichroism (XMCD) measurements for $UCu_2Si_2$ and $UMn_2Si_2$ performed at temperatures of 10 K and 300 K, respectively. In **c** are shown the corresponding ab initio simulations for both compounds. A small difference in the integrated area under the main XANES peak (so called white line) is observed between the two compounds, indicating $UCu_2Si_2$ presents a higher density of unoccupied states above the Fermi level than $UMn_2Si_2$, which may be an indication of the former being much less hybridized than the latter in agreement with the theoretical predictions[25]. The difference observed on XMCD experimental data for the ratio between the intensities of quadrupolar and dipolar intensities is very well reproduced in the simulated spectra and the explanation for this behavior is discussed in the main text. The individual contributions to the theoretical XMCD spectra are shown in **d** for the $UMn_2Si_2$ compound and in **e** for the $UCu_2Si_2$. Interestingly, the amplitude ratio between the two XMCD peaks is opposite for the two compounds. As supported by the calculations of the dipolar/quadrupolar contributions shown in **d** and **e**, this difference is due to the fact that the dipolar contribution is almost the same for both compounds while the quadrupolar contribution has opposite sign between the two materials. Therefore, it is evident that the different $5f$-$6d$ degree of hybridization between these two compounds, defining their Fermi level position, is responsible for the relative alignment between $5f$ and $6d$ moments, which is also consistent with LDA + U simulations used to estimate the $5f$/$6d$ moments (Supplementary Note 3)

such as Curium with $5f^7$ electronic configuration (without orbital moment) could show some quadrupolar contributions if a charge transfer between the $5f$ and $6d$ is promoted in the system. This would be in a similar fashion to what happens to $4f^7$ europium materials when subjected to applied pressures[11], in strong difference to $4f^6$ configuration $Eu_2O_3$ which already shows a strong quadrupolar contribution at the $L_3$ at ambient pressure due to its large orbital moment and spin–orbit coupling[27]. It is also noteworthy that while the $5f$ component in the XMCD spectrum is a contribution mainly from the orbital moment[20], the other terms ($6d$ and $6d$-$5f$) involve more evenly both orbital and spin moments. The availability of a thorough theoretical description of sum rules analysis for this case that could be easily applied to the experimental data without relying on first principles calculations would be an important asset in order to carefully determine the spin and orbital moments of all the contributions in the spectra. Probing the pure $6d$, pure $5f$, and hybridized $6d$-$5f$ terms through dipolar and quadrupolar contributions to the $L_3$-edge XMCD spectrum makes this a unique tool to directly probe both $5f$ and $6d$ orbitals and their hybridization in the same experiment.

**Temperature dependence of the 5f and 6d orbital contributions**. In addition to this orbital-selective probe of the spin-dependent empty density of states using XMCD at the U-$L_3$ edge, it is also worth commenting on the element selectivity of the technique which allowed us to determine that the net uranium moment for both compounds seem to have comparable magnitudes. This is surprising considering the lack of other examples of a magnetic moment on uranium at ambient temperature, as we observed here for the $UMn_2Si_2$. Since a previous neutron scattering study[26] shows ordering of the uranium sublattice only below 100 K, the sizeable XMCD signal in the U atoms at 300 K should be induced by $f$-$d$ hybridization with the magnetically ordered Mn sublattice. This observation of induced quadrupolar contribution is quite unusual especially when compared to the case of rare earths[28]. In $4f$ systems an induced $5d$ moment would not lead to an induced $4f$ moment since these orbitals would not strongly hybridize considering the localized nature of the $4f$ orbitals. Here, on the other hand, the $5f$-$6d$ orbitals hybridize strongly so an induced $6d$ moment can also induce a $5f$ moment. In this scenario we would expect that at low temperatures ($T$<100 K) the XMCD signal should increase due to the magnetic ordering of the uranium sublattice. Indeed, as shown in Fig. 2a–c

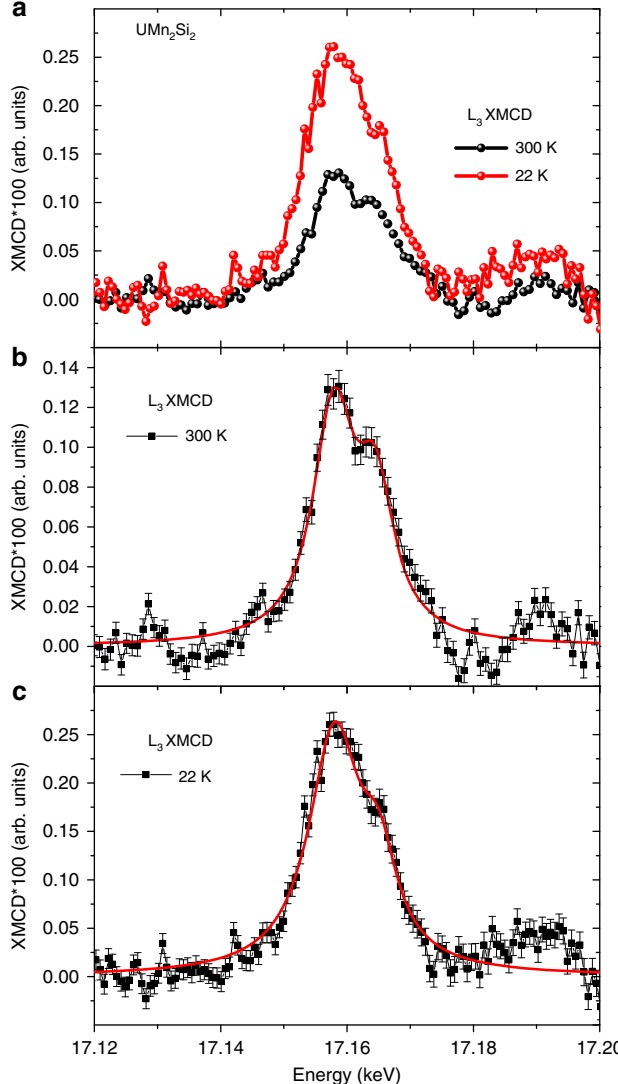

**Fig. 2** Temperature dependence of the X-ray Magnetic Circular Dichroism (XMCD) contributions. **a** XMCD uranium $L_3$ edge for the $UMn_2Si_2$ compound at temperatures of 300 K and 22 K. The experimental XMCD data with the respective statistical error bars for $T = 300$ K and $T = 22$ K are shown in **b**, **c**. The red lines are guides for the eyes to facilitate the visualization of the disproportional changes on the amplitude of each peak (dipolar and quadrupolar) when the temperature is reduced from 300 K to 22 K. Experimental error bars for each energy point are defined as the standard deviation (s.d.) between the multiple averaged spectra

the uranium XMCD signal is overall enhanced at low temperature. Besides this overall increase, the ratio between the intensities of the low energy and higher energy peak is also modified with the temperature reduction. It is clear from the experimental data that the amplitude of the high-energy peak, which is purely due to the $6d$ electron (dipolar) contribution, increases much less than the low energy peak, which has both $5f$ (quadrupolar) and $5f$-$6d$ (hybridized) contributions, when we compare the XMCD data for these two temperatures. This stronger increase at the low energy peak indicates that the $5f$ electrons contribution is enhanced at low temperature much more than the contribution from the $6d$ electrons. This further confirms our assignment of the strong quadrupolar contribution to the first peak of the XMCD at the $L_3$ edge of uranium. This results is also in good agreement with the expected magnetic ordering of the $5f$ electrons as observed by neutron diffraction[26].

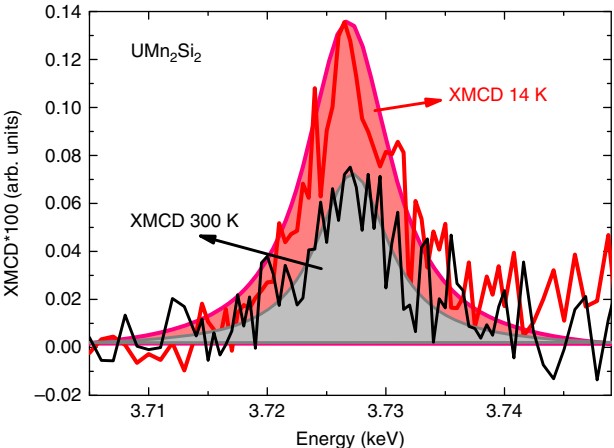

**Fig. 3** Isolated probe of the $5f$ contribution. Uranium $M_4$-edge X-ray Magnetic Circular Dichroism (XMCD) spectra measured for the $UMn_2Si_2$ compound at temperatures of 14 K and 300 K. The area of the XMCD spectra is proportional to the magnetic moment solely of the $5f$ orbitals. By the ratio between the area of the two XMCD spectra we estimate that the uranium $5f$ magnetic moment increases by a factor 2.2, in good agreement with the results obtained at uranium $L_3$ edge (see main text). Note that due to the direct ($3d{\rightarrow}5f$) transition in the dipolar channel at the $M_{4,5}$ the signal amplitude here is a factor of 50 higher than at the $L_3$ edge data as shown in Fig. 2

In order to further confirm the interpretation of our results, we also performed XMCD experiments at the uranium $M_4$ edge (Fig. 3), since at the $M_4$ edge only contribution from the $5f$ orbitals is measured allowing direct verification of the increase in U-$5f$ ordered magnetic moment. The increase in $M_4$-edge XMCD signal by around a factor 2.2 is in excellent agreement with the obtained at low energy peak (quadrupolar) of the $L_3$ edge. This provides an unquestionable evidence of the validity and powerfulness of L-edge XMCD as a tool to simultaneously and selectively investigate, in the same experiment, the contributions from $5f$, $6d$ states and the degree of their hybridization.

Moreover, these temperature dependent results illustrate the relevance of the $5f$-$6d$ hybridization for the magnetic properties of this class of systems. This is also supported by the projected density of states simulations provided in the Supplementary Fig. 2. The $5f$-$6d$ hybridization result in the magnetically ordered Mn $3d$ to induce a moment at both $5f$ and $6d$ uranium orbitals through the overlapped $3d$-$6d$ bands. This leads to our observation of dipolar and quadrupolar contributions to the XMCD signal of the uranium sublattice at room temperature in the $UMn_2Si_2$ compound. This $5f$-$6d$ hybridization acts in the opposite direction in the $UCu_2Si_2$ compound by mediating an induced magnetic moment at the Cu sublattice. This fact is also supported by the observation of a XMCD signal at K-edge of Cu with comparable amplitude to the signal of the Mn (Supplementary Fig. 3). We note that this observation is not a proof that Cu ions are magnetic, but indicate that a considerable amount of uranium $5f$ orbital magnetic moment is transferred to the $4p$-$3d$ Cu orbitals.

In summary, we demonstrate that U-$L_3$ XMCD measurements provide a unique measure of both $5f$ and $6d$ orbitals, as well as their hybridization, through dipolar and quadrupolar contributions to the XMCD spectra. Since this hybridization is key in determining the physical properties of a vast number of actinide compounds this methodology is bound to improve our understanding of these correlated electron systems. To demonstrate the potentiality of our methodology we have studied the magnetic properties of $UCu_2Si_2$ and $UMn_2Si_2$ compounds. The

interpretation of XMCD experiments at uranium $L_3$ edge, together with its temperature dependence, allowed us to demonstrate that the 5f-6d hybridization is the most relevant parameter to determine the magnetic properties of these materials. This hybridization is responsible to induce a magnetic moment at room temperature at the uranium site for the $UMn_2Si_2$ compound as well as to induce a magnetic moment at the Cu sublattice for the $UCu_2Si_2$ compound at low temperature. Furthermore, for the former compound an intrinsic ordering at low temperature was probed by an enhancement of the pure 5f contribution. Considering that the interpretation of the data and conclusions does not rely on theoretical calculations, since the assignment of the quadrupolar/dipolar contributions can be directly tested through temperature dependent experiments and analysis, this methodology provide a unique route to validate theoretical models. It is noteworthy that the methodology developed here, using penetrating high-energy X-rays, can also be applied to study the evolution of f-d hybridization under applied pressure providing a unique tool for tuning electron correlations. The ability to directly probe experimentally the 5f-6d hybridization should guide efforts to understand the magnetic and electronic structures that govern a vast number of actinide compounds with yet bewildering physical properties.

## Methods

**XMCD experiments**. Uranium $L_3$ XMCD measurements were performed at the D04-DXAS beamline of the Brazilian Synchrotron Light Laboratory (LNLS). To cover the energy range around 17.166 keV ($L_3$ edge) we used the (311) and (333) reflections of the bent Si crystal in the dispersive geometry[29]. In this scheme the energy resolution of ~1.3 eV at the $L_3$ edge was set mainly by the area detector used and not by the intrinsic resolution of the high order Bragg reflection. Circularly polarized X-rays were selected by a slit above the orbit plane where the photon flux intensity was 1/3 of the central beam produced by the bending magnet, giving us a degree of circular light of ~54% at the $L_3$ edge[30]. XMCD spectra where obtained from the difference between normalized XANES spectra measured in transmission geometry with opposite directions of the 1 T applied magnetic field. Uranium $M_4$-edge XMCD measurements were performed at the 4-ID-D beamline of the Advanced Photon Source (APS/Argonne)[31] using a 50 μm phase-retarding optics to convert the linear polarization of undulator radiation to circular. $M_4$-edge XMCD spectra were obtained in an equivalent procedure as described above, but switching the helicity of the incoming X-rays in addition to switching the direction of the 0.1 T applied magnetic field which was enough to magnetize the sample. While at the U-$L_3$ edge data were collected on powders in transmission geometry at $T = 300$ and 22 K for $UMn_2Si_2$ and $T = 300$ K for $UCu_2Si_2$, at the $M_4$-edge the data were collected in fluorescence mode. The XMCD signals were corrected for each degree of circular polarization.

**XMCD/XANES calculations**. XANES and XMCD ab initio simulations were performed using the full multiple scattering approach implemented in the FDMNES code[32] including spin–orbit coupling and a 3.0 eV core hole broadening. The atomic potentials obtained from DFT calculations were used as basis and FDMNES code was employed only to include the absorption transition matrix elements to determine the most reliable XANES/XMCD simulations and to avoid difficulties regarding to the Fermi level determination which is critical to XMCD. Since the magnetic moment of both compounds are strongly anisotropic, our simulations were done for the easy magnetization axis of the crystal structure while the experiments were done in a powder sample.

**Density functional theory calculations**. Density functional theory ab initio calculations were performed using the WIEN2K implementation of the full-potential linearized augmented plane-wave method with a double-counting scheme and the rotationally invariant local density approximation LDA + U functional with $U = 1.25$ eV and including spin–orbit coupling[33] for the easy axis magnetization directions as reported in the literature[34, 35]. More detailed parameters for these simulations have already been reported in the literature[36]. Although density functional theory methods are normally not applicable when mixed/fluctuating valence is present, for the systems discussed in this manuscript, however, DFT predictions are indeed validated by experiments which are consistent with a pre-dominant single valence state for uranium in these compounds as strongly supported by the experimental evidence.

**Sample growth**. Polycrystalline samples of $UMn_2Si_2$ and $UCu_2Si_2$ were prepared by arc melting the high purity elements ($U = 99.9\%$, Mn, Cu and Si = 99.9999%) in argon atmosphere followed by an annealing at 800 °C for 5 days in an evacuated

quartz ampoule. The X-ray diffraction pattern confirms the single-phase formation of the compounds. The evaluated lattice parameters and unit cell volume obtained using a Rietveld profile fit are shown in the Supplementary Note 1.

**Data availability**. All relevant data supporting the findings of this study are available from the corresponding author on request. Experimental $L_3$-edge XMCD data for $UMn_2Si_2$ at ambient and low temperature are provided in Supplementary Tables 2 and 3.

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

## Acknowledgements

We are thankful to Roberto Caciuffo and Gerry Lander for discussions and comments on the manuscript. We thank Jose Carlos Botelho Monteiro for providing one of the UMn$_2$Si$_2$ samples. Work at Argonne is supported by the U.S. Department of Energy, Office of Science, Office of Basic Energy Sciences under Contract No. DE-AC-02-06CH11357. Work at LNLS is supported by the Brazilian ministry of science and technology. This research was supported by FAPESP grants 2013/22436-5, 2014/05480-3, 10/19979-9, and 2014/26620-8. R.D.d.R. thanks the funding for his Ph.D. fellowship from CAPES brazilian agency.

## Author contributions

N.M.S.-N. proposed this research. R.D.d.R. and F.G.G. prepared the samples. R.D.d.R., L.S.I.V., and N.M.S.-N. performed U-L$_3$ edge experiments. R.D.d.R., C.A.E., J.C.L., and D.H. performed the M$_4$-edge experiments. R.D.d.R., L.S.I.V., Y.J., and N.M.S.-N. performed the simulations. R.D.d.R., J.C.L., Y.J., D.H., and N.M.S.-N. interpreted the results. R.D.d.R., D.H., and N.M.S.-N. wrote the manuscript. All authors contributed discussing the results and helping to write the final version of the manuscript.
