## [Peer Review File · Nature Communications]

Reviewers' comments:

Reviewer #1 (Remarks to the Author):

This paper presents a novel experimental method to probe 5f-6d hybridization in uranium compounds. The authors proposed and demonstrated measurement of an XMCD spectrum at U-L3 edge (2p → 6d, 5f) instead of conventional M4,5 (3d → 5f) or N4,5 (4d → 5f) edges. The L3 edge has an advantage to probe 6d and 5f at the same time via dipolar and quadrupolar contributions, respectively. The authors succeeded in this measurement in polycrystalline UCu₂Si₂ and UMn₂Si₂ in spite of plausible weak signals from the quadrupole contribution. This would be an important achievement in the experiment. However, as the paper focuses just on technical aspects and does not provide any scientific insight, I do not recommend publishing this article in Nature Communications in the present form. I suggest that the authors try to revise the manuscript by considering the following comments.

- (1) The present manuscript describes just a demonstration of the measurement of XMCD spectra, which include some information about 5f-6d hybridization. The authors should discuss quantitative analysis of the data to deduce materials properties, in particular, UCu₂Si₂ vs. UMn₂Si₂.
- (2) It is not clear how 5f-3d interaction between U and Mn (or Cu) contribute to 5f-6d hybridization in U. The description of the texts in page 2 is very much confusing.
- (3) In Figure 1, I do not understand why the authors show 300-K spectrum for UMn₂Si₂, while 10 K for UCu₂Si₂. They could show at least 22-K spectrum of UMn₂Si₂ as in Figure 2.
- (4) In Figures 1 and 2, the authors should analyze orbital and spin magnetic moments from the XMCD spectra. In particular, which moment is important in the 5f-6d hybridization?
- (5) Although the authors claim that the method is applicable to all actinides compounds, it seems from the text (line 25, left column, page 3) that the large 5f quadrupolar contribution is due to the specific nature of Uranium. Other actinides should be mentioned as well.

Reviewer #2 (Remarks to the Author):

When seeking insights into actinide bonding and electronic structure, scientists typically use L3-edge X-ray absorption spectroscopy. In contrast, X-ray absorption spectroscopies at the N5,4-edge and M5,4-edges are not commonly employed because these edges do not provide information that can be readily interpreted within a molecular orbital or band structure model. Unfortunately, for X-ray magnetic circular dichroism (XMCD), only the N5,4-edge and M5,4-edge XMCD measurements are possible due to limitations in synchrotron instrumentation and theory.

The article provided by Souza-Neto and coworkers shows that these challenges could be overcome and that L3-edge XMCD can be performed for two prototypical uranium compounds, UCu₂Si₂ and UMn₂Si₂. The XMCD measurement shows how to deconvolute transitions associated with final states derived from 5f-6d hybridization and from exclusively 6d final states. The results are consistent with previous experiments (cited by the authors) showing that the 5f-electrons in UCu₂Si₂ are more localized, and that the 5f-electrons in UMn₂Si₂ are more delocalized due to 5f and 6d orbital hybridization. To the best of my knowledge, the results constitute the first measurement of any actinide L3-edge XMCD spectrum. The authors also briefly described a theoretical study to model the L3-edge spectra, which is novel in its own right. The experimental and theoretical work appears to be

of good quality although some details are curious (see below).

This paper is likely to be of very high interest to physicists and chemists who are developing new spectroscopic and theoretical tools to understand actinide electronic structure, magnetism, and physical properties. The paper is also likely to be of broad interest to the much larger community of lanthanide and actinide scientists who are interested in understanding and controlling the relative role of the f and d orbitals - this is currently a hot topic in the field. I am inclined to recommend publication although I have some considerable reservations about the following:

i) Address the role of the other elements in these compounds (Si, Mn, Cu). Is it possible that the changes at the L3-edge are due to changes in the amount of (for example) hybridization between the U 6d and Si 3p? Perhaps the theoretical density of states can shed some light on this question, but it is not discussed in great detail or clearly pictured in the supplemental.

ii) A limitation to this study is the small number of compounds. It would be more convincing if more measurements could be compared. For example, there are a number of centrosymmetric molecular actinide compounds such as $\text{Cs}_2\text{UO}_2\text{Cl}_4$. In this system, 5f and 6d orbital hybridization is forbidden by symmetry (see Denning, Green and coworkers, *J Chem Phys*, 2002, p 8008-8020).

iii) The agreement between theory and experiment is impressive, however the theoretical aspect of this paper is not sufficiently transparent and cannot be adequately reviewed. It's gratifying that the theory and experiment get the trend of UMn_2Si_2 vs UCu_2Si_2 correct, however, there is a 50:50 chance for that outcome. More discussion of the DOS (currently in the SI) should be provided. In general, modeling of L-edge spectra using DFT is extremely challenging, especially when periodic boundary conditions must be applied, and there is little literature precedent for this sort of work.

iv) Some explanation is needed for why the data should be trusted given the large amount of noise, and for why higher quality data could not be obtained. Experimental data showing the full spectral range for the L3-edge XANES and XMCD spectra must be supplied as a table in the supplemental material. Because of the poor data quality, I do not find the fits in Figure 2 to be valid. The datasets for both temperatures have different intensities overall but otherwise have the same spectral profile and are superimposable. If the authors disagree, then error bars for the fit functions should be reported and the experimental data should be supplied in the supplemental material.

In addition, I recommend the following corrections:

- 1) In figure 1c, it is difficult to understand what the arrows are pointing to.
- 2) Throughout the paper, uranium is capitalized.
- 3) First paragraph in "...debate because the 5f orbital is intermediate..." should be "5f electrons are" instead of "5f orbital is" because only electrons can be localized/delocalized.

Reviewer #3 (Remarks to the Author):

This is good work that addresses a current and important problem - that of the electronic structure of actinide materials - which is complicated, and controversial, for all the reasons laid out in the introduction to the manuscript. The authors apply XMCD to address specific questions in valence orbital hybridisation in two uranium materials. They show that they can pull apart the different contributions to the XMCD signal and the paper is ultimately about the power of this technique rather than the materials themselves. It would help the reader if they could be a little clearer on how these

experiments are different from previous L2,3-edge XMCD studies (which they cite), i.e. why are they doing better than previous studies?

I have two other minor queries, which I think could be addressed to help the non-expert reader:

(i) they say they see "The observation of pure 6d, pure 5f, and hybridized 6d-5f through dipolar and quadrupolar contributions to the L3-edge XMCD...". Is this true? You see a spectrum that has contributions from all these terms, and modelling determines these different contributions to the intensity, but I'm not sure you actually directly observe the different contributions?

(ii) the variable temperature data in Figure 2 is used to support the conclusions. But wouldn't you expect an XMCD signal to increase with decreasing temperature anyway?

I am happy to recommend publication after minor revisions.

We thank the Referees for their positive remarks on our manuscript and for their detailed comments. We are grateful that all three of them explicitly acknowledge the importance of our work: Referee 1 judges our data to be an “important achievement”, Referee 2 notes that our work would have “very high interest to physicists and chemists” and Referee 3 considers our work “a very important work” that “reports a key finding.” Additionally, their insightful comments have led us to make revisions that certainly improved the quality of our manuscript. The technical remarks/questions were all addressed on basis of additional experimental and theoretical results as detailed below.

We are confident that our work is a unique and novel experimental effort providing a better understanding of actinides compounds. Therefore, we feel that the current version of the manuscript is very well suited for the audience of Nature Communications and hereby resubmit the manuscript for your consideration.

We include point-by-point replies (in black) to all comments by Referees (*in blue italic*), as detailed below.

Reviewer #1:

This paper presents a novel experimental method to probe 5f-6d hybridization in uranium compounds. The authors proposed and demonstrated measurement of an XMCD spectrum at U-L3 edge (2p -> 6d, 5f) instead of conventional M4,5 (3d -> 5f) or N4,5 (4d -> 5f) edges. The L3 edge has an advantage to probe 6d and 5f at the same time via dipolar and quadrupolar contributions, respectively. The authors succeeded in this measurement in polycrystalline UCu₂Si₂ and UMn₂Si₂ in spite of plausible weak signals from the quadrupole contribution. This would be an important achievement in the experiment. However, as the paper focuses just on technical aspects and does not provide any scientific insight, I do not recommend publishing this article in Nature Communications in the present form. I suggest that the authors try to revise the manuscript by considering the following comments.

We thank the Referee for his/her careful review of our manuscript, and appreciate the positive comment about the importance of our results. We address the constructive comments and questions in the following.

(1) The present manuscript describes just a demonstration of the measurement of XMCD spectra, which include some information about 5f-6d hybridization. The authors should discuss quantitative analysis of the data to deduce materials properties, in particular, UCu₂Si₂ vs. UMn₂Si₂. Following this and similar remarks by the other referees, we have added more details in the manuscript and in the following responses to the referees about the effect of hybridization and other interactions in these two compounds. The determination of orbital and spin moment is also discussed below.

Regarding the quantitative analysis of the data, we agree that it was superficially presented in the initial version of the manuscript. More information about the data fitting is given in the response to item iv of referee #2, which has also been added to the Methods section of the current version of the manuscript.

(2) It is not clear how 5f-3d interaction between U and Mn (or Cu) contribute to 5f-6d hybridization in U. The description of the texts in page 2 is very much confusing.

We thank the referee for pointing out the difficulty to understand the role played by the 5f-3d hybridization. The questions raised stimulated us to clarify several concepts that were most likely explained too tersely in the previous version of the manuscript.

Although the manuscript focuses primarily on the uniqueness of U L-edge spectroscopy to probe 5f, 6d and f-d hybridization via dipolar and quadrupolar interactions, we also provide additional important insight about the two studied samples. For example, the strong hybridization between the uranium 5f and manganese 3d orbitals is responsible for the magnetic response of uranium ions at room temperature. Furthermore, the different relative alignment of 5f and 6d uranium moments in the two compounds is also influenced by hybridization with 3d orbitals. To clarify this discussion, we have added more information about the link between the spectroscopy data and the magnetic properties of both compounds by including, in the supplementary materials, XMCD measurements at the K-edge of Mn and Cu for the two compounds as shown in figure 1 below.

Figure 1: Normalized XANES signal across the Mn and Cu K edges on UT_2Si_2 ($T=Mn, Cu$) samples are presented in black curves. The XANES spectra were normalized to unity at energies well above the edges. Corresponding normalized XMCD data for $\mu_0H = 1$ T measured at $T = 10$ K for Cu edge and at $T = 300$ K for Mn edge are also shown.

In the two systems presented in this work, the most important interaction is the hybridization between the U-6d with the 3d orbitals of Mn or Cu, as evidenced in the projected density of states simulations provided in the updated figure 2 of the supplementary materials (following a recommendation by referee #2). This hybridization result in the magnetically ordered Mn sub-lattice inducing a moment at uranium sites leads to our observation of XMCD signal from the uranium sublattice at room temperature in the UMn_2Si_2

compound. The Mn 3d orbitals polarize the 6d and 5f orbitals of uranium through 3d-6d-5f hybridization (as two peaks are observed in the XMCD dipolar channel). This hybridization is also present in the UCu_2Si_2 compound as evidenced by the observation of a XMCD signal at K-edge of Cu with comparable amplitude to the signal of the Mn (see figure 1 of this document above and the figure 2 of the supplementary materials). We note that this observation is not a proof that Cu ions are magnetic, but indicate that one considerable amount of orbital moment is transferred to the 4p-3d Cu orbitals.

(3) In Figure 1, I do not understand why the authors show 300-K spectrum for UMn_2Si_2 , while 10 K for UCu_2Si_2 . They could show at least 22-K spectrum of UMn_2Si_2 as in Figure 2.

The 300 K spectrum for UMn_2Si_2 together with the 10 K spectrum for UCu_2Si_2 in the figure 1 were collected during the same beamtime, under the same experimental conditions (except for the temperature). These temperatures are far below their respective ferromagnetic ordering temperatures. The measurements at 300 K lead us to the discovery of induced magnetic moment at uranium sites in the Mn compound. After interpreting the results as described in the manuscript it became clear that we should expect the quadrupolar contribution corresponding to the UMn_2Si_2 compound to grow at low temperature if the previous neutron diffraction study [see ref 25 in the manuscript] was to be correct. For that reason, we needed to experimentally confirm our interpretation with the low temperature experiment. Only at a later time we performed the experiment for UMn_2Si_2 below the temperature at which the uranium sublattice should order, as reported by neutron diffraction. Indeed, these later experiments presented in figure 2 completely agree with the initial results and the interpretation of the first peak of the XMCD to be a contribution of the 5f (and 5f-6d) in nature. Considering that this seems to be the first ever reported L_3 -edge XMCD on an actinide material (see below), this result and robustness of the interpretation is to us the most important achievement of this work. This opens a plethora of opportunities to study the electronic and magnetic structure of actinide compounds which strongly relies on the 5f-6d interaction.

(4) In Figures 1 and 2, the authors should analyze orbital and spin magnetic moments from the XMCD spectra. In particular, which moment is important in the 5f-6d hybridization?

The appearance of the quadrupolar contribution at the L_3 -edge is specially related to a large orbital moment of the system, while the 5f-6d hybridization spectral contribution includes more evenly information on both orbital and spin moments.

The use of magneto-optical sum rules to separately determine the orbital and spin magnetic moments from XMCD spectra as proposed by P. Carra et al [Phys. Rev. Lett. 70, 694 (1993)] is a unique tool. However, these sum rules normally cannot be directly applied for the case of L-edges of rare earths and actinides due to the strong influence of the f -states. For the case of actinide materials, normally M-edges are used for this end since it only probes the 5f moments [see references 19 and 20 of the manuscript]. While some ways to overcome this have been proposed [eg.: Europhys. Lett. 66 441 (2004)], it relies on the inclusion of spin-asymmetry parameters and the combination of the experimental data with theoretical calculations, in addition to the correct extraction of the 5f quadrupolar contribution from the spectra.

We note that to be able to determine the orbital and spin moments using the sum rules, we would have to measure the XMCD spectra at the uranium L_2 -edge at energy of 21 keV, for which we would have a factor of 10x less photon flux, at the current Brazilian synchrotron source, together with a 20% decrease in circular polarization as compared to the uranium L_3 -edge. Other beamlines at higher energy synchrotron sources would probably have difficulties to obtain circular polarization at this energy as well. These

experimental limitations make the attempt to measure the L₂-edge XMCD of uranium very challenging and time consuming, even more than the already difficult experiments at the L₃-edge (each L₃-edge XMCD took 3 days of continuous data collection to achieve good counting statistics).

In addition, for uranium compounds that do present a very strong spin-orbit-coupling as seen by M-edge experiments [19,20], and therefore a strong orbital magnetic moment, the quadrupolar contribution will be almost negligible at the L₂-edge. This is because while the quadrupolar and octupolar terms are more important at high energies, when looking at systems with large spin-orbit-coupling the quadrupolar contribution to absorption cross section has a different factor depending if applied to L₂ or L₃ edge. While the quantitative theory behind this is explained elsewhere (Galera, Joly et al [ref. 22 in our manuscript]), we show here below experimental and theoretical XMCD data on Eu₂O₃ as a rare earth analogy to our study on uranium compounds (N. M. Souza- Neto et al [ref 26 in our manuscript]). The differences between L₃ and L₂ edges highlight the strong spin-orbit coupling that forces the large quadrupolar contribution (2p → 4f) to appear almost only at the L₃ edge. This is also supported by the ab initio simulations done with FDMNES code which accurately predicts the position and shape of the spin-up and spin-down quadrupolar (4f) contribution. In the case we present here, we feel that our proof-of-concept is robust because it does not depend on the simulations to get to the final conclusions and these experimental results can be used to validate theoretical models. Therefore, even if we had performed the challenging L₂-edge XMCD measurements experiment for the UMn₂Si₂ compound, we could not trust results from sum rules analysis due to the strong reliance on the theoretical description of the spatial extent of 5f orbitals, spin-asymmetry parameter, and a robust definition of a theoretical procedure in order to apply sum rules in this case.

On the other hand, since our experimental data show both dipolar (6d) and quadrupolar (5f) and their hybridized (5f-6d) contributions in the same spectrum, it is noteworthy that while the 5f term is a contribution mainly from the orbital moment of the compound [ref. 15 in our manuscript], the other terms (6d and 6d-5f) involves move evenly both orbital and spin moment. The availability of a thorough theoretical description of sum rules analysis for this case that could be easily applied to the experimental data without relying on first principles calculations would be an important asset in order to carefully determine the spin and orbital moments of all the contributions in the spectra.

Figure 2: Europium $L_{2,3}$ -edges XMCD on Eu_2O_3 high purity samples. The experimental and theoretical data are compared showing a remarkable agreement. [N. M. Souza- Neto et al [ref 26 in our manuscript]]

(5) Although the authors claim that the method is applicable to all actinides compounds, it seems from the text (line 25, left column, page 3) that the large $5f$ quadrupolar contribution is due to the specific nature of Uranium. Other actinides should be mentioned as well.

We appreciate these comments from the referee. We clarified this wording in the main text as follows:

“(…) the high energy of the actinides L-edges (16 to 25 keV) enhances the quadrupolar contribution from the transition Hamiltonian operator when compared to lower energy L-edges of rare earths[11, 21, 26] (5 to 10 keV)”

As pointed out by referee #2: “To the best of my knowledge, the results constitute the first measurement of any actinide L_3 -edge XMCD spectrum”. We believe that referee #2 is correct and no other L_3 -edge XMCD on actinide materials has been reported so far. Considering this, we can claim that this method can be applied to any other actinide compound with a disproportion between spin-up and spin-down density of states (either intrinsic to the $5f$ -orbitals, induced by neighboring ions or by very large applied magnetic fields).

In addition to that, we can make an analogy to the previous reports of observed quadrupolar contributions in the rare earths (see Galera, Joly et al [ref. 22 in our manuscript]). The spin-orbit-coupling and presence or not of a sizeable orbital moment plays a crucial role on the appearance of quadrupolar contribution. Moreover, the high energy of the uranium L-edges (17 keV) enhances the quadrupolar contribution from the transition Hamiltonian operator when compared to lower energy L-edges of rare earths[refs: 11, 21, 26] (5 to 10 keV) as described in the manuscript. In that sense, we can argue that all magnetic actinide materials would present L_3 -edge XMCD spectra including quadrupolar contributions. In addition to that, even systems without orbital moment, such as Curium with $5f^7$ electronic configuration, could show some quadrupolar contributions if a charge transfer between the $5f$ and $6d$ is promoted in the system in similar

fashion to what happens to $4f^7$ europium materials when subjected to applied pressures (see N.M. Souza-Neto et al [ref 11 in our manuscript]), in strong difference to $4f^6$ configuration europium compounds with strong orbital moment as presented in the figure 2 above (see N. M. Souza- Neto et al [ref 26 in our manuscript]).

Following this recommendation of the referee, we added the following in the manuscript to address the applicability of this method in other actinide materials:

“In addition, considering that spin-orbit-coupling and presence of a sizeable orbital moment plays a crucial role on the appearance of a quadrupolar contribution, we can envision that all magnetic actinide materials may present sizeable quadrupolar contributions in their L_3 -edge XMCD spectra. Even materials such as Curium with $5f^7$ electronic configuration (without orbital moment) could show some quadrupolar contribution if a charge transfer between the 5f and 6d is promoted in the system. This would be in a similar fashion to what happens to $4f^7$ europium materials when subjected to applied pressures (see N.M. Souza-Neto et al [ref 11 in our manuscript]), in strong difference to $4f^6$ configuration Eu_2O_3 which already shows a strong quadrupolar contribution at the L_3 at ambient pressure due to its large orbital moment and spin-orbit-coupling [see N. M. Souza- Neto et al [ref 26 in our manuscript]].”

Reviewer #2:

When seeking insights into actinide bonding and electronic structure, scientists typically use L_3 -edge X-ray absorption spectroscopy. In contrast, X-ray absorption spectroscopies at the $N_{5,4}$ -edge and $M_{5,4}$ -edges are not commonly employed because these edges do not provide information that can be readily interpreted within a molecular orbital or band structure model. Unfortunately, for X-ray magnetic circular dichroism (XMCD), only the $N_{5,4}$ -edge and $M_{5,4}$ -edge XMCD measurements are possible due to limitations in synchrotron instrumentation and theory

The article provided by Souza- Neto and coworkers shows that these challenges could be overcome and that L_3 -edge XMCD can be performed for two prototypical uranium compounds, UCu_2Si_2 and UMn_2Si_2 . The XMCD measurement shows how to deconvolute transitions associated with final states derived from 5f-6d hybridization and from exclusively 6d final states. The results are consistent with previous experiments (cited by the authors) showing that the 5f-electrons in UCu_2Si_2 are more localized, and that the 5f-electrons in UMn_2Si_2 are more delocalized due to 5f and 6d orbital hybridization. To the best of my knowledge, the results constitute the first measurement of any actinide L_3 -edge XMCD spectrum. The authors also briefly described a theoretical study to model the L_3 -edge spectra, which is novel in its own right. The experimental and theoretical work appears to be of good quality although some details are curious (see below).

This paper is likely to be of very high interest to physicists and chemists who are developing new spectroscopic and theoretical tools to understand actinide electronic structure, magnetism, and physical properties. The paper is also likely to be of broad interest to the much larger community of lanthanide and actinide scientists who are interested in understanding and controlling the relative role of the f and d orbitals - this is currently a hot topic in the field. I am inclined to recommend publication although I have some considerable reservations about the following:

We thank the Referee for his/her careful review of our manuscript, and appreciate the positive comment about the importance of our results. We clarify the constructive comments and questions in the following.

i) Address the role of the other elements in these compounds (Si, Mn, Cu). Is it possible that the changes at the L3-edge are due to changes in the amount of (for example) hybridization between the U 6d and Si 3p? Perhaps the theoretical density of states can shed some light on this question, but it is not discussed in great detail or clearly pictured in the supplemental.

We thank the referee for raising this question. Following the suggestion of the referee, we carefully looked again at our simulations and changed figure 2 of the supplementary material to make easier the visualization of hybridization region (the new figure is also shown below). Indeed, as previously mentioned in the manuscript, the hybridization is a key parameter for the interpretation of magnetic properties in uranium compounds. In the two systems presented in this work, the more important interaction is the hybridization between the U-6d with the 3d orbitals of Mn or Cu. This hybridization is the main responsible for our observation of a XMCD signal at the uranium sublattice at room temperature for the UMn_2Si_2 compound due to the induced moment at the uranium site by the magnetically ordered Mn sub-lattice. The Mn 3d orbitals polarize the 6d and 5f orbitals of uranium through 3d-6d-5f hybridization (as two peaks are observed in the XMCD dipolar channel). This hybridization is also present on UCu_2Si_2 compound as evidenced by the observation of a XMCD signal at the K-edge of Cu with comparable amplitude to the signal of the Mn (see figure 1 of this document above and the figure 2 of the supplementary materials). We note that this observation is not a proof that Cu ions are magnetic, but indicate that one considerable amount of orbital moment is transferred to the 4p-3d Cu orbitals.

Regarding the Si atom, as the referee suggests, looking at our simulations the hybridization of Si orbitals with those of others elements of the material is almost the same for both compounds. Besides, the induced magnetic moment in the Si is about one hundred times lower than the magnetic moment for the uranium. This insight provided by the theoretical simulations make us to believe that, at least for these compounds, the magnetic contribution from the Si atom can be completely neglected in the interpretation of the changes of the L₃-edge XMCD signal. We added the following in the supplementary materials to address this:

“These simulations also shed some light regarding the contribution from interactions with Silicon orbitals. The hybridization of Si orbitals with those of others elements is almost the same for both compounds. However, the induced magnetic moment in the Si atoms is about one hundred times lower than the magnetic moment observed for the uranium. This make us to believe that, for at least these compounds, the magnetic contribution from the Si atom can be completely neglected in the interpretation of the changes of the L₃-edge XMCD signal.”

Figure 3 UMn_2Si_2 and UCu_2Si_2 simulations of the dipolar and quadrupolar contributions to the L₃-edge XMCD determined using the FDMNES code, as well as the density of states determined by LDA+U calculations

*ii) A limitation to this study is the small number of compounds. It would be more convincing if more measurements could be compared. For example, there are a number of centrosymmetric molecular actinide compounds such as $Cs_2UO_2Cl_4$. In this system, 5f and 6d orbital hybridization is forbidden by symmetry (see Denning, Green and coworkers, *J Chem Phys*, 2002, p 8008-8020).*

In fact, we have performed L₃-edge XMCD experiments on a few other uranium compounds, as presented below. Although the main feature is reproduced in all the compounds, these other uranium compounds have different crystalline structure and several unusual properties, which makes them not ideal to exemplify the applicability of the technique since there is not a simple and direct model and comparison for all of them in view of electronic effects.

Figure 4. Experimental uranium L_3 edge XANES and XMCD measurements for UTe, UGe₂ and UGa₂. The data for UTe was taken at both APS and LNLs. To allow an easier comparison the amplitude was also normalized by the ratio of each respective saturation magnetic moment to the moment of UGa₂.

In addition to the performed XMCD measurements for several uranium compounds, in item (4) of response to referee #1 we argue about the validity of this method for other actinide materials. Here we ended up deciding to only present the results for UCu₂Si₂ and U Mn₂Si₂ because they are isostructural compounds with temperature dependent properties and open questions on the actinide magnetic sub-lattice and hybridization with neighboring ions. This makes them ideal prototypical systems to apply this new approach and make a straightforward comparison of their electronic properties. Our focus on these compounds is also justified by our motivation to present one central story in order to clearly demonstrate this approach and maximize the impact and perspective for future works.

We thank the referee for the suggestion to study this centrosymmetric compound. Indeed, it would be very interesting to perform uranium L_3 -edge XMCD in a compound where the $5f$ - $6d$ hybridization is forbidden, since then we could probe the isolated quadrupolar contribution. However, this approach is based on XMCD measurements which require the sample to present a net disproportionality between spin-up and spin-down density of unoccupied states. This is found on ferromagnetic materials after being magnetized by magnetic fields higher than the coercive field and for anti-ferromagnetic and paramagnetic materials in applied fields strong enough to align the spins along the field direction. To the best of our knowledge this centrosymmetric Cs₂UO₂Cl₄ compound does not present any form of magnetic order, therefore it cannot be studied with this approach. Nevertheless, perhaps an extremely high magnetic field (>30 T) [Phys. Rev. Lett. 103, 046402 (2009)] may be able to induce a net magnetization in this material, although strongly altering the ground state electronic structure by the large amount of magnetic energy introduced.

iii) The agreement between theory and experiment is impressive, however the theoretical aspect of this paper is not sufficiently transparent and cannot be adequately reviewed. It's gratifying that the theory and experiment get the trend of U Mn₂Si₂ vs UCu₂Si₂ correct, however, there is a 50:50 chance for that outcome. More discussion of the DOS (currently in the SI) should be provided. In general, modeling of L-edge spectra using DFT is extremely challenging, especially when periodic boundary conditions must be applied, and there is little literature precedent for this sort of work.

We agree with the referee in his assessment that “*modeling of L-edge spectra using DFT is extremely challenging*”. More than that, we can add that it is challenging for LDA+U methods to correctly capture the electronic structure of actinide materials where the 5f “band” is somewhere between localized

electrons such as the 4f orbitals in rare earths and the delocalized 3d orbitals of transition metals. That motivated us to concentrate the foundation of this manuscript on the experimental aspects which serves to validate the theoretical results. We felt fortunate that these theoretical methods worked quite well for these cases we are reporting. Below we give more information on how it was employed.

The XMCD simulations showed in the main text were performed using the LDA+U approximation only to calculate the initial electronic potential while the XANES/XMCD spectra were obtained using the multiple scattering approach employed in the FDMNES code. Similar results were also obtained using only the FDMNES to fully simulate the XMCD spectra without having the initial input from LDA+U calculations. On top of that, to exemplify what would happen with the XMCD spectra if the electronic 5f/6d configurations were different from the one discussed in this work, we have performed several simulations for the UMn_2Si_2 compound considering an electronic configuration with and without net moment in the 5f U orbital. As can be seen from the plot in the Figure below, if there was no moment in the 5f states, the dipolar (6d) contribution by far does not resemble the experimental data. On the other hand, the configuration with a moment in the U 5f orbital does present a quadrupolar contribution and resembles the experimental data. All these simulations below were performed without the LDA+U potentials from DFT calculations (as in the simulations presented in the manuscript) in order to force this hypothetical electronic configuration. That supports our argument that the only scenario where theory agrees with the experiment is when we do have a moment on the 5f orbitals which reflects on the XMCD amplitude through the quadrupolar (pure 5f) contribution and hybridized 6d-5f term present in the dipolar contribution.

Figure 4: XMCD simulations using the FDMNES code for hypothetical configurations with sizeable moment on the 5f orbitals (left) or zero moment on the 5f (right).

iv) Some explanation is needed for why the data should be trusted given the large amount of noise, and for why higher quality data could not be obtained. Experimental data showing the full spectral range for the L3-edge XANES and XMCD spectra must be supplied as a table in the supplemental material. Because of the poor data quality, I do not find the fits in Figure 2 to be valid. The datasets for both temperatures have different intensities overall but otherwise have the same spectral profile and are superimposable. If the authors disagree, then error bars for the fit functions should be reported and the experimental data should be supplied in the supplemental material.

We thank the referee for this suggestion. We agree that further information on the fitting shown in figure 2 must be provided. We have included the tabled data in the supplemental material as recommended.

The fitting was performed using Lorentzian functions initially fixing the peak energy and width to the values observed experimentally, while putting no constrain on the fitted peak amplitude. Then we removed the constrains to obtain the final fitting of the parameters. The fitting of the data resulted in a “reduced χ ” of 0.953 and “adjusted R^2 ” of 0.978 for the UMn_2Si_2 dataset at low temperature, with similar numbers for the other spectral fitting. Taking into account the experimental error bar and the convergence criteria we obtained an error bar for each fitted parameter of about 5%. Considering the propagation of uncertainties of the two independent datasets, we ended up having the following ratio between the two temperatures for the dipolar, quadrupolar and hybridized contributions: $5f_{ratio} = 2.288 \pm 0.272$; $6d_{ratio} = 1.334 \pm 0.120$; $5f-6d_{ratio} = 1.329 \pm 0.119$. These uncertainties are in very good agreement with the discussion provided in the main text of the manuscript, since this final uncertainty is enough to unquestionably conclude that the quadrupolar contribution (5f) at low temperature increases significantly more than the other dipolar contributions. As discussed in the main text, this agrees with the uranium lattice in UMn_2Si_2 being fully ordered at low temperature but displaying smaller induced magnetization at room temperature as a result of hybridization with Mn 3d orbitals.

While figure 2 simply mentions the factors 2.3 and 1.3, we have added the complete previous paragraph to the experimental methods section at the end of the current version of the manuscript.

The data quality is basically in the limit of what we can experimentally achieve nowadays. The measurements of each final XMCD spectrum took approximately 3 days (72 hours) of continuously accumulation to achieve the required counting statistics by averaging more than 3000 spectra, at the our current Brazilian synchrotron source. The error bar supplied in figure 2 reflects the standard deviation among all measured spectra. Considering that normally one experiment at a synchrotron facility takes no more than one week per semester, all spectra shown here (together with measurements in other compounds shown above) represent the result of a long effort to have a robust and concise proof of concept on this method. With experimental optimizations and developments currently taking place and the advent of 4th generation synchrotron sources (such as the SIRIUS Brazilian synchrotron source under construction, Advanced Photon Source upgrade, European Synchrotron Radiation Facility upgrade, among others), we expect that these experiments will require much less effort to achieve better signal/noise ratios in the very near future, opening up opportunities for a broad range of studies in magnetic and electronic properties of actinide materials.

In addition, I recommend the following corrections:

1) In figure 1c, it is difficult to understand what the arrows are pointing to.

2) Throughout the paper, uranium is capitalized.

3) First paragraph in “...debate because the 5f orbital is intermediate...” should be “5f electrons are” instead of “5f orbital is” because only electrons can be localized/delocalized.

We thank the referee for these corrections. All the suggested revisions were implemented in this new version of the manuscript.

Reviewer #3 (Remarks to the Author):

This is good work that addresses a current and important problem - that of the electronic structure of actinide materials - which is complicated, and controversial, for all the reasons laid out in the introduction to the manuscript. The authors apply XMCD to address specific questions in valence orbital hybridization in two uranium materials. They show that they can pull apart the different contributions to the XMCD signal and the paper is ultimately about the power of this technique rather than the materials themselves. It would help the reader if they could be a little clearer on how these experiments are different from previous L_{2,3}-edge XMCD studies (which they cite), i.e. why are they doing better than previous studies?

We thank the referee for his/her good assessment of our work and comments below.

Regarding the comment “*how these experiments are different from previous L_{2,3}-edge XMCD studies (which they cite)*”, as we comment in the reply to referee #1 and pointed out by referee #2, to the best of our knowledge, no other L₃-edge XMCD on actinide materials has been reported so far. The reports we cite are for L_{2,3}-edge XMCD studies done for rare earth materials, and here we use them as analog studies to facilitate our presentation and interpretation. The difficulty to perform uranium L₃-edge XMCD is described in the main text, involving a low XMCD signal amplitude, difficulty to have a large degree of circularly polarization, low photon flux in the energy range, and the issues related to the handling of the “radioactive” samples due to safety regulations at synchrotron facilities.

I have two other minor queries, which I think could be addressed to help the non-expert reader: (i) they say they see "The observation of pure 6d, pure 5f, and hybridized 6d-5f through dipolar and quadrupolar contributions to the L₃-edge XMCD...". Is this true? You see a spectrum that has contributions from all these terms, and modelling determines these different contributions to the intensity, but I'm not sure you actually directly observe the different contributions?

In our results, we observe a peak that is purely due to the dipolar contribution (6d orbitals) and another that is a combination of quadrupolar (pure 5f) and dipolar (hybridized 5f-6d states and pure 6d) contributions. This is supported by our interpretation, by temperature dependent experiments (figure 2), from M₄-edge experiments (figure 3) and theoretical simulations of the XMCD spectra and their isolated contributions. In this way, we confirm that we are indeed probing the three contributions in the two peaks of the XMCD spectra. As pointed by the referee, we agree that to distinguish between the 5f and 5f-6d hybridization in the low energy XMCD peak, it is helpful to have the support from theory (as described in the manuscript) or from the XMCD at M₄ edge (as in the figure 3 of the manuscript) which probes only the 5f contribution. Following the referee comment, we have changed this sentence to “The probe of pure 6d, pure 5f, and hybridized 6d-5f ...”.

(ii) the variable temperature data in Figure 2 is used to support the conclusions. But wouldn't you expect an XMCD signal to increase with decreasing temperature anyway?

We agree with the referee that we would expect an increase of the XMCD signal when we decrease temperature, since the magnetic moment of the sample would also increase. However, for the U_mN₂Si₂ compound the observed room temperature XMCD signal is a result of an induced magnetic moment due

to the strong hybridization between the uranium and Manganese orbitals. According to a previous neutron diffraction report (reference 25 in the manuscript), below 100 K the uranium lattice orders magnetically, and then we should expect a larger increase of the 5f contribution compared to the 6d contribution. Indeed, we verify that changing the temperature from 300 K to 20 K results in a sizable change of the (5f) quadrupolar contribution (low energy peak) but a much smaller change in the dipolar contribution. To come to this conclusion, we fitted the XMCD peaks considering Lorentzian functions for all orbital contributions (5f, 6d, 5f-6d) at different temperatures. We found from the fitting that while the quadrupolar contribution in the low temperature data increases by a factor of ~ 2.3 related to the ambient temperature, the dipolar contribution increases just by a factor of ~ 1.3 , as also detailed in the reply to the item (iv) of referee #2. To verify the validity of the fitting we measured XMCD at uranium M₄ edge, which only give us information about the 5f moment. The experiment at M₄ edge found that the 5f magnetic moment increases by a factor of 2.2, in astonishing agreement with our observation at L₃ edge. All these results confirm the validity of our new methodology to investigate magnetic properties of actinide compounds.

I am happy to recommend publication after minor revisions.

We thank the referee for taking the time to review our manuscript, the constructive comments and for the recommendation of acceptance for publication.

Reviewers' comments:

Reviewer #1 (Remarks to the Author):

The authors tried to revise the manuscript according to the comments. However, they should further consider the following points.

- (1) Although the authors described details of experiment and interpretation in their reply to reviewers, the revision of the manuscript is limited only to small parts. They should include many of the details in the manuscript as well.
- (2) The authors did not reply to the suggestion that the manuscript should provide scientific insight rather than just focus on technical aspects. They should revise the abstract and the summary sections to describe what was found for UCu_2Si_2 and UMn_2Si_2 from a point of view of materials properties.

Reviewer #2 (Remarks to the Author):

The revised manuscript and supporting information provided by Souza-Neto and coworkers satisfactorily addresses many of the reviewer comments. These are challenging measurements to make and the data is very difficult to interpret. With anticipated improvements to synchrotron beamline intensity these experiments are likely to become more common, and in that regard this manuscript is breaking very new ground.

However I am not satisfied with the authors' response to my comment regarding the validity of the fits in Figure 2. As written, it seems as though the interpretation is being used to support the fits and not the other way around. The 22K spectrum is clearly more intense overall, but otherwise the 20K and 300K spectral profiles are virtually identical. When the 20K data is multiplied by 0.5, both spectra are almost superimposable. Granted, there is a slight difference near 17.16 keV, however, this is small difference in a few data points relative to the large statistical error of the measurement. I am also wary of the statement in the Figure 2 caption "Experimental error bars are shown only for the low temperature experiments in (c), while the other measurements share similar level of statistics." Since these are very new types of measurements, more work is needed to demonstrate that other reasonable models of the data are invalid.

I would be satisfied if the authors' chose to forgo reporting fitting results and instead reported a qualitative analysis based on the overall change in spectral intensity, which could be due to the increase in $5f$ contribution. At this stage it seems that alternative explanations including experiment error cannot be ruled out.

Reviewer #3 (Remarks to the Author):

The authors have addressed the queries I raised. I think the manuscript reads much better now and in particular the introduction is much clearer. I recommend publication.

Reply to referees:

We thank the reviewers for their positive comments and valuable suggestions. Below we reply to each of their comments in detail. We have modified the manuscript accordingly. All changes are indicated in the answers for the referees as well as highlighted in the article file. We are convinced that our revisions attend all the open question from the reviewers and that the revised version of the manuscript is now suitable for publication in Nature Communications.

Reviewer #1 (Remarks to the Author):

The authors tried to revise the manuscript according to the comments. However, they should further consider the following points.

(1) Although the authors described details of experiment and interpretation in their reply to reviewers, the revision of the manuscript is limited only to small parts. They should include many of the details in the manuscript as well.

(2) The authors did not reply to the suggestion that the manuscript should provide scientific insight rather than just focus on technical aspects. They should revise the abstract and the summary sections to describe what was found for UCu_2Si_2 and UMn_2Si_2 from a point of view of materials properties.

We thank the referee for his/her recommendations. We include several points from the previous reply letter into the manuscript. We also modified both abstract and summary section to describe what was found for both compounds. The changes on the manuscript are shown below:

Abstract: We included the sentence: “We found that while in the UCu_2Si_2 compound the 5f-6d hybridization conveys a magnetic moment at the copper sublattice, for the UMn_2Si_2 compound this hybridization mediates, in the opposite way, an induced uranium magnetic moment at room temperature. Furthermore, for the latter compound an intrinsic ordering at low temperature was probed by an enhancement of the pure 5f contribution.”

Main text: We included the following paragraphs to explain the validity of the sum rules in 4f/5f materials:

Page 2, 1st column:

“We note that the use of magneto-optical sum rules to separately determine the orbital and spin magnetic moments from XMCD spectra as proposed by P. Carra et al [ref 23] is

a unique tool. However, these sum rules normally cannot be directly applied for the case of L-edges of rare earths and actinides due to the presence of 5f quadrupolar contributions in the XMCD spectra and its influence on the spatial extent of 6d orbitals. While some ways to overcome this have been proposed [ref 24], it relies on the inclusion of spin-asymmetry parameters and the combination of the experimental data with theoretical calculations, in addition to the correct extraction of the 5f quadrupolar contribution from the spectra. The actual models for sum rules for 5f materials requires XMCD measurements in both L3 and L2 edges which with the limitations on flux and polarization of the current Synchrotron sources are still very challenging and time consuming experiments. However, with anticipated improvements to synchrotron brightness these experiments are likely to become more common.”

Page 3, 2nd column:

“It is also noteworthy that while the 5f component in the XMCD spectrum is a contribution mainly from the orbital moment[22], the other terms (6d and 6d-5f) involve more evenly both orbital and spin moments. The availability of a thorough theoretical description of sum rules analysis for this case that could be easily applied to the experimental data without relying on first principles calculations would be an important asset in order to carefully determine the spin and orbital moments of all the contributions in the spectra. Probing the pure 6d, pure 5f and hybridized 6d-5f terms through dipolar and quadrupolar contributions to the L 3 -edge XMCD spectrum makes this a unique tool to directly probe both 5f and 6d orbitals and their hybridization in the same experiment.”

We also include one paragraph discussing the relevance of the hybridization between U and Cu/Mn for the magnetic properties of the two studied compounds (page 4, 2nd column):

“Moreover, these temperature dependent results illustrate the relevance of the 5f-6d hybridization for the magnetic properties of this class of systems. This is also supported by the projected density of states simulations provided in figure 2 of the supplementary materials. The 5f-6d hybridization result in the magnetically ordered Mn 3d to induce a moment at both uranium orbitals through the overlapped 3d-6d bands. This leads to our observation of dipolar and quadrupolar contributions to the XMCD signal of the uranium sublattice at room temperature in the UMn_2Si_2 compound. This 5f-6d hybridization acts in the opposite direction in the UCu_2Si_2 compound by mediating an induced magnetic moment at the Cu sublattice. This fact is also supported by the observation of a XMCD signal at K-edge of Cu with comparable amplitude to the signal of the Mn (see figure 3 of the supplementary materials). We note that this observation is not a proof that Cu ions are magnetic, but indicate that one considerable amount of uranium 5f orbital magnetic moment is transferred to the 4p-3d Cu orbitals. “

Summary: We added one sentence highlighting which information from the samples was learnt with our new methodology:

“To demonstrate the potentiality of our new methodology we have studied the magnetic properties of UCu_2Si_2 and UMn_2Si_2 compounds. The interpretation of XMCD experiments at uranium L_3 edge, together with its temperature dependence, allowed us to demonstrate that the 5f-6d hybridization is the most relevant parameter to determine the magnetic properties of these materials. This hybridization is responsible to induce a magnetic moment at room temperature at uranium site for the UMn_2Si_2 compound as well as to induce a magnetic moment at the Cu sublattice for the UCu_2Si_2 .”

Reviewer #2 (Remarks to the Author):

The revised manuscript and supporting information provided by Souza-Neto and coworkers satisfactorily addresses many of the reviewer comments. These are challenging measurements to make and the data is very difficult to interpret. With anticipated improvements to synchrotron beamline intensity these experiments are likely to become more common, and in that regard this manuscript is breaking very new ground.

We thank the referee for recognizing the difficult of our measurements. Indeed, we expect that with improvements of the synchrotron facilities these challenging experiments, as demonstrated in our work, will become more usual.

However I am not satisfied with the authors' response to my comment regarding the validity of the fits in Figure 2. As written, it seems as though the interpretation is being used to support the fits and not the other way around. The 22K spectrum is clearly more intense overall, but otherwise the 20K and 300K spectral profiles are virtually identical. When the 20K data is multiplied by 0.5, both spectra are almost superimposable. Granted, there is a slight difference near 17.16 keV, however, this is small difference in a few data points relative to the large statistical error of the measurement. I am also wary of the statement in the Figure 2 caption “Experimental error bars are shown only for the low temperature experiments in (c), while the other measurements share similar level of statistics.” Since these are very new types of measurements, more work is needed to demonstrate that other reasonable models of the data are invalid.

We now include the statistical error bars for the XMCD dataset at 300 K in the figure 2 following the point raised by the referee.

I would be satisfied if the authors' chose to forgo reporting fitting results and instead reported a qualitative analysis based on the overall change in spectral intensity, which could be due to the increase in 5f contribution. At this stage it seems that alternative explanations including experiment error cannot be ruled out. *We follow the referee recommendation and removed the fits from figure 2. As pointed*

out by the referee, we can describe the results qualitatively, describing the physical phenomena of the sample and the potential of our new methodology without using the data fitting. The interpretation of the results remain the same, as the qualitative analysis of the experimental data come to the same conclusions as in the previously described quantitative data fitting. Below we present the new figure 2 and their respective discussion.

Figure: (a) XMCD uranium L_3 edge for the UMn_2Si_2 compound at temperatures of 300 K and 22 K. The experimental XMCD data with the respective statistical errors bars for $T = 300$ K and $T = 22$ K are shown in (b) and (c). The red lines are guides for the eyes to facilitate the visualization of the disproportional changes on the amplitude of each peak (dipolar and quadrupolar) when the temperature is reduced from 300 K to 22 K.

Page 4:

“Indeed, as shown in figure 2(a) the uranium XMCD signal is overall enhanced at low temperature. Besides this overall increase, the ratio between the intensities of the low energy and higher energy peak is also modified with the temperature reduction. It is clear from the experimental data that the amplitude of the high energy peak, which is purely due to the 6d electron (dipolar) contribution, increases much less than the low energy

peak, which has both 5f (quadrupolar) and 5f-6d (hybridized) contributions, when we compare the XMCD data for these two temperatures. This stronger increase at the low energy peak indicates that the 5f electrons contribution is enhanced at low temperature much more than the contribution from the 6d electrons. This further confirms our assignment of the strong quadrupolar contribution to the first peak of the XMCD at the L_3 edge of uranium. This results is also in good agreement with the expected magnetic ordering of the 5f electrons as observed by neutron diffraction[ref 25].

In order to further confirm the interpretation of our results, we also performed XMCD experiments at the uranium M_4 edge (see figure 3), since at the M_4 edge only contribution from the 5f orbitals is measured allowing direct verification of the increase in U-5f ordered magnetic moment. The increase in M_4 -edge XMCD signal by around a factor 2.2 is in excellent agreement with the obtained at low energy peak (quadrupolar) of the L_3 edge. This provides an unquestionable evidence of the validity and powerfulness of L-edge XMCD as a tool to simultaneously and selectively investigate, in the same experiment, the contributions from 5f, 6d states and the degree of their hybridization.

Reviewer #3 (Remarks to the Author):

The authors have addressed the queries I raised. I think the manuscript reads much better now and in particular the introduction is much clearer. I recommend publication.

We thank the referee for taking the time to review our manuscript and for the recommendation for publication.

REVIEWERS' COMMENTS:

Reviewer #1 (Remarks to the Author):

The authors revised the manuscript so as to describe the role of 5f-6d hybridization in UMn_2Si_2 and UCu_2Si_2 clearly. The difference in the magnetic properties of the two compounds is now well written, which is informative not only in X-ray spectroscopy but also in materials research. I recommend that the manuscript be published.

Reviewer #2 (Remarks to the Author):

This reviewer provided confidential remarks recommending publication.